# Intracellular Sugar Transporters Facilitate Cellulase Synthesis in *Trichoderma reesei* Using Lactose

**DOI:** 10.3390/biom13020295

**Published:** 2023-02-04

**Authors:** Haiyan Wang, Ai-Ping Pang, Bingzhi Li, Liujie Huo, Fu-Gen Wu, Fengming Lin

**Affiliations:** 1State Key Laboratory of Food Nutrition and Safety, Tianjin University of Science & Technology, Tianjin 300453, China; 2State Key Laboratory of Bioelectronics, School of Biological Science and Medical Engineering, Southeast University, Nanjing 210096, China; 3Key Laboratory of Systems Bioengineering (Ministry of Education), School of Chemical Engineering and Technology, Tianjin University, Tianjin 300072, China; 4Helmholtz International Laboratory, State Key Laboratory of Microbial Technology, Shandong University, Qingdao 266237, China

**Keywords:** sugar transporter, lactose, lignocellulose-degrading enzyme

## Abstract

Sugar transporters play an important role in the cellulase production of lignocellulose-degrading fungi. Nevertheless, the role and function of these transporters are still unclear. Here we first report intracellular sugar transporters assisting cellulase production in *Trichoderma reesei* (*T. reesei*) using lactose. The mRNA levels of sugar transporter genes *mfs*, *gst*, and *lac1* were substantially upregulated in *T. reesei* cultivated on lactose, with the most abundant mRNA levels at 24 h as compared to glucose. Moreover, the individual deletion of these sugar transporters significantly inhibited cellulase production, solid cell growth, and sporulation of *T. reesei*, suggesting they play a supporting role in cellulase production when grown in lactose. Surprisingly, MFS, GST, and LAC1 were mainly localized in the cytoplasm, with MFS and LAC1 in the endoplasmic reticulum (ER), representing the first discovery of intracellular sugar transporters involved in cellulase biosynthesis in lactose culture. The absence of the gene *lac1* noticeably inhibited most of the crucial genes related to cellulase production, including cellulase-encoding genes, transcription factors, and sugar transporters, at 24 h, which was fully relieved at 48 h or 72 h, indicating that *lac1* affects cellulase production more at the early step. This research advances the understanding of the function of intracellular sugar transporters in fungi, particularly for fungal cellulase production.

## 1. Introduction

*Trichoderma reesei* (*T. reesei*) decays wood in nature, metabolizing the cellulose and hemicellulose of the plant biomass via cellulases and hemicellulases. Cellulase is a complicated, extracellular enzyme mixture, majorly comprised of three kinds of synergistic enzymes for cellulose degradation: endoglucanase (EG, EC3.2.1.4), exoglucanase (or cellobiohydrolases, CBH, EC 3.2.1.91), and β-glucosidase (BGL, EC 3.2.1.21) [1]. Cellulase is produced by microbial fermentation for applications in the biofuel and biorefinery industries using *T. reesei* RUT-C30 as the working horse. Lactose, the byproduct from the whey processing and cheese manufacturing industries, is inexpensive and economically feasible. Despite the fact that it is not as efficient as cellulose in cellulase induction, lactose is the only major soluble carbon source for cellulolytic enzyme fermentation by *T. reesei* on an industrial level [2]. Nevertheless, the mechanism behind the induction of cellulase by lactose is not understood, limiting the metabolic engineering efforts for improved cellulase induction efficiency of lactose.

Sugar transporters have crucial functions in sensing and transporting diverse sugars for cellulase production in lignocellulose-degrading fungi, such as Crt1 and Stp1 in *T. reesei* [3,4,5], cellodextrin transporters CDT-1, CDT-2, and CLP1 in *Neurospora crassa* [6,7], cellobiose transporters *cltA* and *cltB* in *Aspergillus nidulans* (*A. nidulans*) [8], and cellodextrin transporters CdtC, CdtD, and CdtG in *Penicillium oxalicum* [9]. In *T. reesei*, there is a large array of sugar transporters that are capable of effectively transporting the constituent sugars into the cell, enabling its access to sugars. In contrast to that, *Saccharomyces cerevisiae* has only 43 sugar transporters, while *T. reesei* has 113 putative sugar transporters in its in silico proteome [10]. Recently, intracellular sugar transporters have been reported for cellulase production in *T. reesei* on cellulose [11,12].

It is assumed that lactose can be hydrolyzed extracellularly by β-galactosidase (lactase, β-D-galactosidase galactohydrolase; E.C 3.2.1.23) to D-glucose and D-galactose, that are then taken up by cells through the respective permeases [13]. Alternatively, lactose itself can be taken up by specific lactose permeases and hydrolyzed intracellularly. The cellular uptake of these sugars is crucial for cell growth and is intricately linked to cellulase and hemicellulase production. Accordingly, sugar transporters are found to play a critical role in cellulase production using lactose. Transporters for cellular internalization of lactose were induced by cellulose-derived sugars [14]. Meanwhile, several key sugar transporters have been identified as involved in cellulase production from lactose. For instance, the lactose transporter Crt1 (Trire2: 3405) appears to be essential for growth on lactose, and its deletion prevents lactose-induced cellulase gene expression in *T. reesei* [3,5,15], which is proved to exist in the plasma membrane and the periphery of the nucleus [11]. A screening for lactose permeases in *T. reesei* PC-3-7 was performed, leading to the identification of two MFS sugar transporters (TR79202 and TR77517) whose deletion decreased lactose uptake, cell growth, and cellulase production on lactose with no significant effect on cellulase production in cellulose culture [4]. This demonstrated that the two identified sugar transporters are important for cellulase generation when using lactose as the carbon source. A lactose permease gene (lacpA) was found in another cellulase-producing fungus, *A. nidulans*, whose deletion impaired lactose uptake and cell growth [16]. Despite the great importance of sugar transporters in cellulase production, the molecular mechanism behind the function and role of sugar transporters in cellulase production remained undetermined.

Here, the function of three sugar transporters, MFS (M419DRAFT_137795), GST (M419DRAFT_138519), and LAC1 (M419DRAFT_127980), was investigated in cellulase production using lactose as the carbon source. We measured the mRNA levels of these sugar transporters and assayed their effect on cellulase activities by knocking them out in *T. reesei*, finding that they facilitate cellulase production. Furthermore, we investigated the impact of *lac1* on the transcriptional levels of cellulase-related genes like cellulase-encoding genes, transcription factors, other sugar transporters, and the lactose metabolism enzyme, revealing that the function of *lac1* in cellulase expression acted at the transcriptional levels at an early stage of cellulase production, which was mediated by the well-known cellulase regulators CRT1 and XYR1. Finally, we analyzed the cellular distribution of sugar transporters MFS, GST, and LAC1 by C-terminally tagging them with the red fluorescent protein DsRed, unraveling that they are localized in the cytoplasm, especially in the endoplasmic reticulum (ER). This study might offer new strategies to improve fungal cellulase production by engineering intracellular sugar transporters.

## 2. Materials and Methods

### 2.1. Strains and Cultivation Conditions

*Escherichia coli* (*E. coli*) DH5α was utilized to construct plasmids, and *Agrobacterium tumefaciens* (*A. tumefaciens*) AGL-1 was used for fungal transformation. Strain *T. reesei* Rut-C30 (CICC 13052, ATCC 56765) was used as a parental strain for gene overexpression, and strain Δku70 was used as a parental strain for gene knockout and in situ fluorescence protein tagging [17,18]. *T. reesei* Δku70, a derivate of *T. reesei* Rut-C30, created by deleting the gene *ku70*, was kindly provided by Professor Wei Wang from East China University of Science and Technology [19]. *E. coli* DH5α and *A. tumefaciens* were grown on Luria-Bertani medium at 37 °C and 28 °C, respectively. *T. reesei* were cultured in Sabouraud Dextrose Broth (SDB) and Trichoderma minimal media (TMM) [20] containing 2% lactose as the sole carbon source at 28 °C for the analysis of cellulase production and gene regulation, and cultured on potato dextrose agar (PDA) plates and TMM solid plates with lactose at 28 °C for the analysis of cell growth and sporulation ability.

### 2.2. Construction of Recombinant T. reesei Strains

To knockout genes *mfs*, *gst*, and *lac1*, the 1.5-kilobase (KB) upstream and downstream fragments of these genes were amplified from the genomic DNA of Δku70 and then integrated into plasmid pXBthg, resulting in plasmids pXBthg-mfs, pXBthg-gst, and pXBthg-lac1. The recombinant plasmids were then transformed into parental strain Δku70, respectively, through the Agrobacterium-mediated transformation (AMT) method [21] to obtain the recombinant strains Δmfs, Δgst, and Δlac1.

For linking a red fluorescent protein DsRed at the C-terminus of the transporter to track its intracellular localization, the upstream (including the promoter and coding sequence) and downstream (including the terminator) fragments of *mfs*, *gst*, and *lac1* amplified from the genomic DNA of Δku70 were ligated to the plasmid pXBred [22], respectively. These plasmids were transformed into the parental strain Δku70 by the AMT method, leading to the recombinant strains MFS-DsRed, GST-DsRed, and LAC1-DsRed, respectively. In addition, the fragment of gene *lac1* was amplified from the genomic DNA of Rut-C30 and inserted into plasmid p-DsRed [22] at the XbaI site under a modified cbh1 promoter, leading to plasmid lac1-pDmred. Plasmid lac1-pDmred was transformed into parental strain Rut-C30 using the AMT method, generating the recombinant strain LAC1-DsRed-OE. All the gene modifications in the recombinant strains were confirmed by polymerase chain reaction (PCR) and sequencing of the PCR products at Sangon Biotech. The primers used for amplification and verification are listed in Appendix A.

### 2.3. The Shake Flask Culture of T. reesei

An amount of 0.5 mL of *T. reesei* conidia (10^7^/mL) was seeded into 10 mL SDB and grown for 48 h at 28 °C with 200 rpm. Following this, 5 mL pre-grown mycelia (the wet weight of mycelia was about 0.6 g) were inoculated into a 250 mL conical flask containing 50 mL TMM media with 2% lactose, and cultured for 72 h at 28°C with 200 rpm. Samples were collected at different time points for cellulase and hemicellulase activity assays [22,23,24,25,26], confocal observation, and quantitative reverse transcription PCR (qRT-PCR) analysis. For analyzing the cell growth in liquid culture medium, 1 mL of a fungal culture solution containing diffuse mycelia was collected at 24 h, 48 h, and 72 h and diluted appropriately, and then the absorption value at 600 nm wavelength of the diluent was measured using an ultraviolet spectrophotometer (UV-2600, Shimadzu, Japan). In order to measure the growth ability of the recombinant strains on the solid medium, 10 μL of fresh spore solution was dropped into the center of the solid TMM plates. Then, the colony diameter was measured at 48 and 72 h, respectively.

### 2.4. qRT-PCR Analysis

Fresh mycelia were preserved at the indicated time points. Total RNA was extracted from mycelia cells using the Fungi RNA Kit (Omega Bio-Tek, Inc., USA, R6840) and then reverse transcribed into cDNA using PrimeScript™ RT reagent Kit with gDNA Eraser (Takara, Dalian, China, RR047Q). qRT-PCR was performed in an ABI 7500 real-time PCR system, following the protocol of TB Green Premix Ex Taq™ II (Takara, Dalian, China, RR820Q). The relative mRNA level of each tested gene was normalized to the housekeeping gene *sar1* [27,28] and was calculated by the 2^−ΔCt^ method, where ΔCt = Ct(test) − Ct(sar1). All the primers used are described in Appendix A. Three biological replicates were performed, with three technical replicates for each biological replicate.

### 2.5. Confocal Imaging

Confocal images of *T. reesei* strains DsRed [29], MFS-DsRed, GST-DsRed, LAC1-DsRed, and LAC1-DsRed-OE cultured on lactose were taken using a confocal laser scanning microscopy (CLSM) TCS SP8 with a 100× oil immersion objective. The 552 nm laser was selected to excite the samples, and the fluorescence emission was detected between 570 and 700 nm. *T. reesei* mycelium was collected at different time points through centrifuging at 8000 rpm for 15 min. For labeling ER, *T. reesei* mycelium was washed twice with Hank’s Balanced Salt Solution (HBSS) buffer and resuspended in 1 mL HBSS. Then ER-Green (KeyGEN BioTECH Co., Ltd., Jiangsu, China) was added into the suspension at a ratio of 1:1000 and incubated at 37 °C for 20 min. After incubation, the culture was washed twice with HBSS and resuspended in HBSS. Then, the images of stained samples were taken with an excitation wavelength of 488 nm and an emission wavelength range of 500–550 nm. For dying cell membranes, *T. reesei* mycelium was washed with Phosphate Buffered Saline (PBS) and then resuspended in staining solution GC-PEG-cholesterol-FITC and incubated in the dark at 37 °C for 10 min [22,30,31]. After incubation, the culture was washed with PBS twice and resuspended in PBS. The images of samples stained with cell membrane dye were taken with an excitation wavelength of 488 nm and an emission wavelength range of 500–550 nm.

### 2.6. Bioinformatic Analysis

To obtain the transmembrane domains of these sugar transporters, the amino acid sequences of these sugar transporters were downloaded from the EnsemblFungi database and then submitted to the web server Protter, respectively. To find the ER localization signal motif of sugar transporters, the amino acid sequences of these sugar transporters were submitted to the CLUSTALW online server, and the outcomes were then submitted to the ESPript online server.

## 3. Results

### 3.1. The Transcription Dynamics of Sugar Transporters MFS, GST, and LAC1 during the Cellulase Production on Lactose

To explore the role of sugar transporters MFS, GST, and LAC1 on cellulase generation, we measured the mRNA levels of these three genes during the whole process of cellulase production on lactose (cellulase-producing condition) by qRT-PCR in comparison to glucose (no cellulase-producing condition) (Figure 1). The transcriptional abundances of these sugar transporters on lactose were notably higher than those on glucose, similar to that of the well-known cellulase-essential sugar transporter gene *crt1*(M419DRAFT_109243). Particularly, the relative mRNA levels of *mfs*, *gst*, and *lac1* were 8.81, 0.33, and 0.01 in *T. reesei* on lactose for 24 h, which were 11.40, 54.90, and 17.11 times that on glucose. Moreover, gene *mfs* possessed the highest mRNA abundance, followed by *gst* and *lac1* in a decreasing order, but was lower than that of *crt1*. This remarkable upregulation in the gene expression of *mfs*, *gst*, and *lac1* under lactose conditions is indicative of their importance for cellulase production when using lactose as the carbon source.

### 3.2. Sugar Transporters MFS, GST, and LAC1 Facilitated Cellulase and Hemicellulase Production in T. reesei Grown on Lactose

Inspired by their notable transcriptional upregulation under lactose conditions, we individually knockout genes encoding putative sugar transporters MFS, GST, and LAC, in the parental strain Δku70 by homologous recombination, yielding strains Δmfs, Δgst, and Δlac1, respectively. The enzyme-producing ability of these deletion strains was evaluated by measuring cellulase and hemicellulase activities in their culture supernatant at varied time points using lactose as the carbon source, as compared to the starting strain Δku70 (Figure 2). The single deletion of sugar transporters MFS, GST, and LAC1, substantially inhibited the cellulase and hemicellulase production in *T. reesei* grown on lactose. When grown on lactose for 72 h, strain Δlac1 exhibited 1.25 IU/mL FPase activity, 4.02 IU/mL CMCase activity, 0.14 IU/mL pNPCase activity, 0.59 IU/mL pNPGase activity, and 0.11 IU/mL pNPxase activity, and a 3.23 mg/mL secreted protein concentration, only 30.54%, 66.81%, 31.24%, 63.68%, 43.44%, and 59.06% of that in strain Δku70, respectively. The other two knockout strains showed similar reduction degree to strain Δlac1. Clearly, the sugar transporters MFS, GST, and LAC1 are crucial for cellulase and hemicellulase production in *T. reesei*.

### 3.3. The Knockout of Sugar Transporters MFS, GST, and LAC1 Reduced the Cell Growth and Spore Ability of T. reesei Cultivated on Lactose

When being cultured in TMM+2% lactose, the cell growth of strains Δmfs and Δgst was noticeably reduced at 24 h and 48 h but increased at 72 h, while strain Δlac1 showed decreased cell growth at all tested time points as compared to Δku70 (Figure 3A). When grown on TMM+2% lactose agar plates, the three deletion strains Δmfs, Δgst, and Δlac1 showed much smaller colony diameters than Δku70, only 86.7%, 84.9%, and 56.7% of that Δku70 at 72 h (Figure 3B,C). The single deletion of these transporter genes, *mfs* (Trire2: 79202), *gst* (Trire2: 50894), or *lac1* (Trire2: 56684), in QM9414 also led to growth reduction using lactose as the carbon source [3]. Meanwhile, all three knockout strains showed a significantly reduced number of spores in comparison to Δku70 (Figure 3B,D). These data demonstrate that the sugar transporters MFS, GST, and LAC1 play a positive role in both cell growth and sporulation with *T. reesei* together in cellulase biosynthesis.

### 3.4. The Impact of lac1 Knockout on the mRNA Levels of Crucial Genes Involved in Cellulase Production

Though the decline of enzyme activities of these three constructed strains Δmfs, Δgst, and Δlac1, was similar, the growth of strain Δlac1 was more significantly repressed when cultured on lactose. Hence, we chose strain mutant Δlac1 to perform qRT-PCR analysis. We monitored the mRNA dynamics of key genes associated with (hemi)cellulase expression in strain Δlac1 cultivated on lactose, including genes encoding cellulase, which include β-glucosidase BGL1 (*cel3a*), the major cellobiohydrolase CBHI (*cel7a*), the major endoglucanase CMC (*cel7b*), hemicellulse β-xylosidase (*bxl1*), the well-known transcription factors *xyr1* [32], *ace3* [33], *clr2* [34], *clr3* [35,36], and *xpp1* [37], sugar transporters *crt1* and *stp1* [5], and gene *xyl1* in the lactose metabolism. Except genes *xpp1* and *stp1*, which stayed unchanged, all the assayed genes were notably downregulated in strain Δlac1 at 24 h, as compared to Δku70. Nevertheless, the mRNA levels of these genes were comparable to or even higher than those in Δku70 at 48 h and/or 72 h (Figure 4). The transcriptional abundance of *xpp1* and *stp1* stayed almost constant in the first 48 h, but was noticeably reduced at 72 h.

### 3.5. Cellular Distribution of Sugar Transporters MFS, GST, and LAC1

Structure analysis showed that these transporters are transmembrane proteins belonging to MFSs and displaying characteristics of sugar transporters. There are 12 transmembrane domains in GST and 11 in MFS and LAC1 (Appendix A). Nevertheless, the cellular distribution of these sugar transporters is still unknown. To investigate the cellular distribution of sugar transporters MFS, GST, and LAC1, we tagged the fluorescence protein DsRed at their C termini, by integrating the DsRed gene sequence at their 3′ end through homologous recombination, obtaining recombinant strains MFS-DsRed, GST-DsRed, and LAC1-DsRed expressing the fusion proteins MFS-DsRed, GST-DsRed, and LAC1-DsRed. CLSM was utilized to track these red-emitting fusion proteins (Figure 5A and Appendix A). Strain MFS-DsRed started to show weak red fluorescence in the cytoplasm at 24 h, which became much stronger at 48 h and 72 h. At 48 h, the protein MFS-DsRed was dominantly localized to characteristic perinuclear ER rings, exhibiting bright red fluorescence. Nevertheless, red fluorescence was also found on the cell membrane, although it was not that significant, indicating that MFS-DsRed can also distribute to the cell membrane. No yellow fluorescence was found in the strain MFS-DsRed co-labelled with a green cell wall dye, GC-PEG-cholesterol-FITC (Appendix A), implying that MFS were on the cell membrane but not the cell wall. The red fluorescence in strain GST-DsRed was only observed at 72 h but not at 24 or 48 h, while no fluorescence was found in strain LAC1-DsRed at all studied time points. Unlike MFS-DsRed which was localized in perinuclear ER rings, dense red fluorescent punctuates were observed in DsRed. At 72 h, the fluorescence intensity in strain MFS-DsRed was the highest, followed by that in strains GST-DsRed and LAC1-DsRed, which was consistent with their transcriptional abundance order.

To visualize the protein LAC1 in *T. reesei*, the overexpression of LAC1-DsRed under the strong promoter cbh1 was carried out, leading to the recombinant strain LAC1-DsRed-OE. Strain LAC1-DsRed-OE displayed red fluorescence in perinuclear ER rings and vacuoles, similar to strain MFS-DsRed. This suggests that LAC1 was distributed in perinuclear ER rings and vacuoles in *T. reesei* grown on lactose. Moreover, the red fluorescence at the apical regions of the recombinant strains MFS-DsRed, GST-DsRed, and LAC1-DsRed-OE was not stronger than that at the subapical areas, showing that all three sugar transporters were not accumulated at the apical regions (Appendix A).

To check whether the sugar transporters are ER-targeted, mutant strains DsRed, MFS-DsRed, GST-DsRed, and LAC1-DsRed-OE were labeled with a green-emissive ER-tracker (Figure 5B). The red fluorescence of MFS-DsRed and LAC1-DsRed-OE was overlapped with the green fluorescence of the ER tracker, leading to the yellow fluorescence in the merged figure, showing that MFS-DsRed and LAC1-DsRed-OE were localized in ER. In contrast, no yellow fluorescence was found in the merged picture derived from the pictures of GST-DsRed and ER tracker, indicating that GST-DsRed was not distributed in the ER. Moreover, no yellow fluorescence was found in strain DsRed, indicating that the fluorescence protein DsRed alone was not distributed in the ER. Bioinformatics analysis showed that there was an ER localization signal KKXX (Lys-Lys-X-X) [38,39] in the amino acid sequence of LAC1 (Appendix A). However, interestingly, there was no such ER localization signal sequence was found in transporters MFS and GST (Appendix A).

## 4. Discussion and Conclusions

Currently, much of the research efforts on cellulase production using lactose as the carbon source has focused on the lactose metabolism [1,40,41,42,43,44,45]. On the contrary, the importance of sugar transporters on cellulase formation by lactose has not received widespread interest, although some efforts have been made. For instance, Ivanova et al. performed a system analysis of the *T. reesei* transcriptome when using lactose as the carbon source and identified a lactose permease that is essential for cellulase induction [3]. Herein, the function and cellular distribution of three predicted sugar transporters (MFS, GST, and LAC1) were investigated under cellulase-generating conditions when using lactose as the carbon source. First, the mRNA levels of these three sugar transporters were much higher in Rut-C30 grown on lactose than those on glucose. Furthermore, cellulase production was much lower in the deletion strains Δmfs, Δgst, and Δlac1 than in the parental strain Ku70. Both these results demonstrate that sugar transporters MFS, GST, and LAC1 facilitate cellulase production using lactose (Figure 6). Several putative lactose permeases have been found to work in cellulase production in lignocellulosic fungi, such as the proposed lactose transporter Crt1 [3,5,15], two MFS sugar transporters (TR79202 and TR77517) [4] in *T. reesei,* and the lactose permease gene (lacpA) in *A. nidulans* [16]. Just like MFS, GST, and LAC1, cellulose response transporter 1 (CRT1) belongs to the MFS transporter and is initially considered a lactose permease. Crt1 is required for cellulase biosynthesis on lactose, cellulose, or sophorose, and is highly expressed in *T. reesei* cultivated in lactose [4], cellulose [4], and wheat straw [46]. Different from, where the single knockout of *crt1* was enough to fully repress cellulase formation in *T. reesei* cultivated on lactose [3,5], the sole deletion of *mfs*, *gst*, and *lac1* severely but not completely repressed cellulose production in *T. reesei*, showing that these sugar transporters play a supporting rather than an essential role in cellulase production. Another possible explanation for this might be that there are other functionally equivalent sugar transporters to MFS, GST, or LAC1 in *T. reesei* that can circumvent the loss of these sugar transporters and allow for cellulase production.

The time-course analysis of the mRNA levels of critical genes for cellulase expression in strain Δlac1 cultured in lactose for cellulase generation revealed that the deletion of LAC1 notably reduced the transcriptional abundances of these genes except *xpp1* and *stp1* at 24 h. The severe reduction of transcriptional levels of cellulase-encoding genes *cel3a*, *cel7a*, and *cel7b* was consistent with the notable inhibition of cellulase activities (Figure 2B–D). Meanwhile, the mRNA levels of the well-known cellulase transcriptional activator XYR1 and sugar transporter CRT1, both of which are essential for cellulase formation, were significantly reduced. Nevertheless, this transcriptional suppression was recovered at 48 h and/or 72 h. It seems that LAC1 works at the transcriptional level at the early stage of cellulase production, which is crucial for the initiation of cellulase production with lactose.

More importantly, the cellular distribution analysis by CLSM showed that MFS, GST, and LAC1 were dominantly localized in the cytoplasm, though a minor fraction of MFS was sometimes was found on the cell membrane (Figure 6). They were distributed evenly throughout the whole hyphal, without accumulation in the apical regions. As far as we know, this is the first report on intracellular sugar transporters for cellulase production induced by lactose. Previous studies focused on the membrane-bound sugar transporters for cellulase production in lignocellulosic fungi, like Crt1 and Stp1 in *T. reesei* [5], cellodextrin transporters CDT-1, CDT-2, and CLP1 in *Neurospora crassa* [6,7], and cellobiose transporters *cltA* and *cltB* in *Aspergillus nidulans* [8]. The localization of MFS, GST, and LAC1 in cytoplasm and ER implies they may function in sugar sensing, transporting, and/or utilization inside the cells after its cellular uptake. Particularly, MFS and LAC1 were majorly distributed to ER (Figure 6), as indicated by a large number of perinuclear ER rings found in strains MFS-DsRed and LAC1-DsRED-OE (Figure 5A), and the red fluorescence of proteins MFS-DsRed and LAC1-DsRED-OE was overlapped with the green fluorescence of the ER tracker (Figure 5B). ER is a well-known place for the glycosylation of proteins and lipids, requiring a regular replenishment of cytosolically synthesized nucleotide sugars, which are transported by a group of dedicated nucleotide sugar transporters (NST) [47,48]. There is a possibility that MFS and LAC1 are involved in protein glycosylation in the ER, acting as nucleotide sugar transporters, endomembrane sugar transporters, or sugar transceptors. Similarly, these three sugar transporters were also localized intracellularly in *T. reesei* grown on cellulose [12]. Moreover, the well-known lactose permease Crt1, which is essential for cellulase production on lactose [15] and had been considered to be on the cell membrane for a long time, was recently found to be intracellular, mostly at the periphery of the nucleus [11]. Nevertheless, the detailed roles of these intracellular sugar transporters remain unknown, which is worth exploring in the future. There is a possibility that the localization of MFS and GST in ER is an artifact of stressful conditions, which we believe is probably not the case in our study. If the localization of MFS and GST in the ER was due to the ER stress, where ER is overloaded with misfolded proteins, it would probably cause the fluorescence quenching of DsRed. Then, no fluorescence would be observed in the ER of strains MFS-DsRed and LAC1-DsRed, which would be contrary to our result (Figure 5). In addition, through a literature search and amino acid sequence analysis, we found that some of the analyzed sugar transporters contained an ER localization signal (Appendix A). There is one ER localization signal sequence reported in the literature: KKXX (Lys-Lys-X-X) [38,39] which existed in LAC1. Interestingly, no ER localization signal sequence was observed in MFS or GST. Nevertheless, further experimentation is required to completely exclude this possibility. For example, protein expression can be performed under ER stress caused by adding drugs like tunicamycin. As the first-ever report on the discovery and function of intracellular sugar transporters for cellulase production in lignocellulosic fungi that we know of, this study deepens our understanding of fungal intracellular sugar transporters and cellulase production mechanisms.

In summary, we reported that sugar transporters MFS and LAC1 were localized in the ER, while GST was in the cytoplasm. They have a positive role in cellulase production and biomass formation in *T. reesei* when cultured in lactose (Figure 6).

## Figures and Tables

**Figure 1 biomolecules-13-00295-f001:**
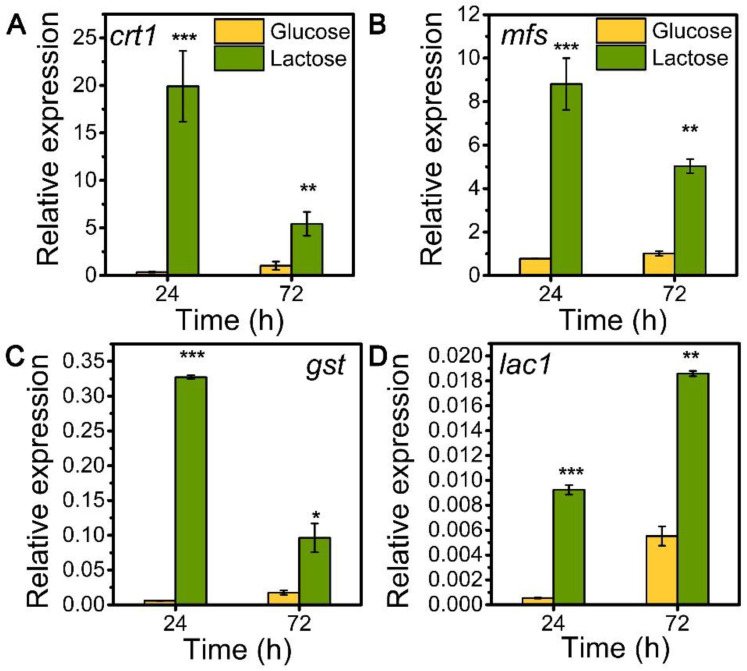
Transcription levels of sugar transporter genes (**A**) *crt1*, (**B**) *mfs*, (**C**) *gst*, and (**D**) *lac1* in *T. reesei* Rut-C30 cultured on TMM+2% lactose/glucose at 24 h and 72 h. Values are the means of three biological replicates, and error bars are the standard deviations of these three replicates. The gene expression level was calculated using the 2^−ΔCt^ method. The relative mRNA level of each tested gene was normalized to the housekeeping gene *sar1*. * *p* < 0.05, ** *p* < 0.01, and *** *p* < 0.001.

**Figure 2 biomolecules-13-00295-f002:**
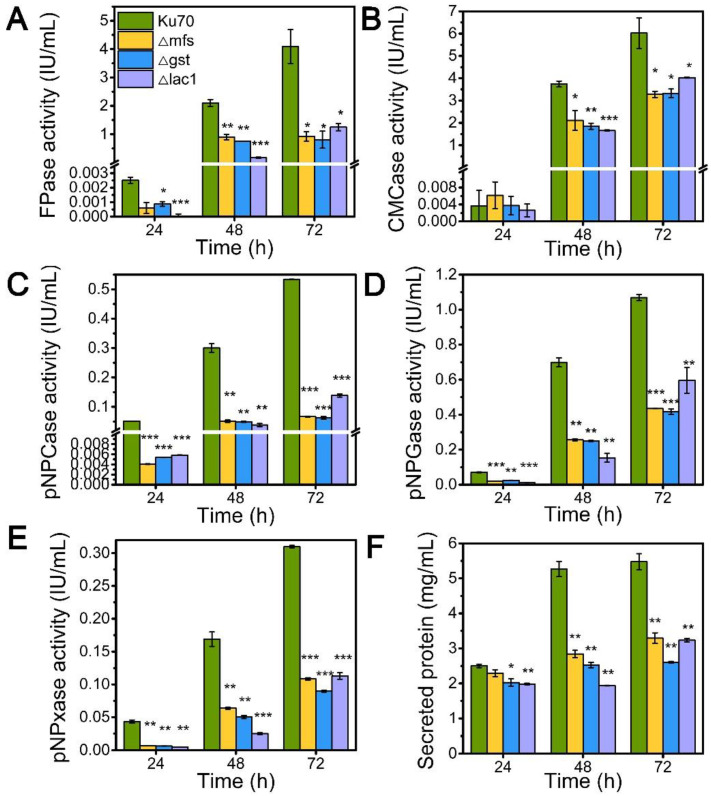
The cellulase and hemicellulase activities and protein secretion of *T. reesei* Δku70, Δmfs, Δgst, and Δlac1 cultured in TMM+2% lactose at 24 h, 48 h, and 72 h, respectively. (**A**) FPase: the filter paper activity; (**B**) CMCase: the CMC activity; (**C**) pNPCase: the CBH activity; (**D**) pNPGase: the BGL activity; (**E**) pNPxase: the xylanse activity; (**F**) protein secretion. Values are the means of three biological replicates, and error bars are the standard deviations of these three replicates. Asterisks indicate significant differences (* *p* < 0.05, ** *p* < 0.01, and *** *p* < 0.001) as assessed by the Student’s *t* test.

**Figure 3 biomolecules-13-00295-f003:**
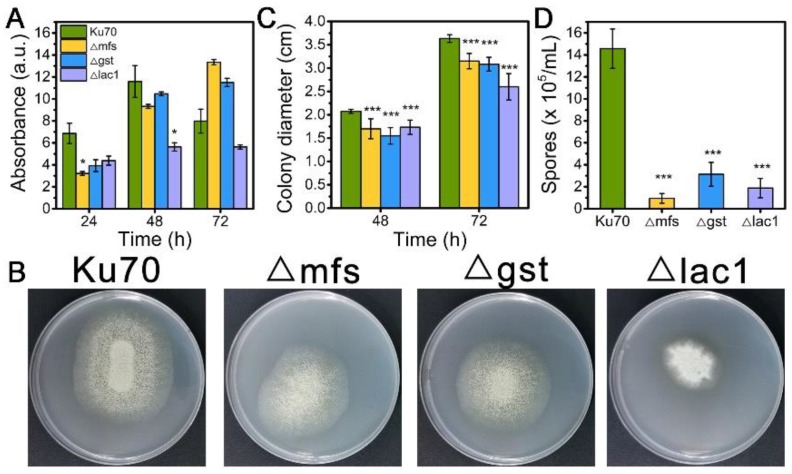
(**A**) The growth, (**B**) plate growth pictures, (**C**) colony diameters, and (**D**) sporulation of *T. reesei* Ku70, Δmfs, Δgst, and Δlac1 grown on TMM with 2% lactose. Both the spore assay and the plate growth pictures were performed at 72 h. Values are the means of three biological replicates, and error bars are the standard deviations of these three replicates. Asterisks indicate significant differences (* *p* < 0.05, and *** *p* < 0.001) as assessed by the Student’s *t* test.

**Figure 4 biomolecules-13-00295-f004:**
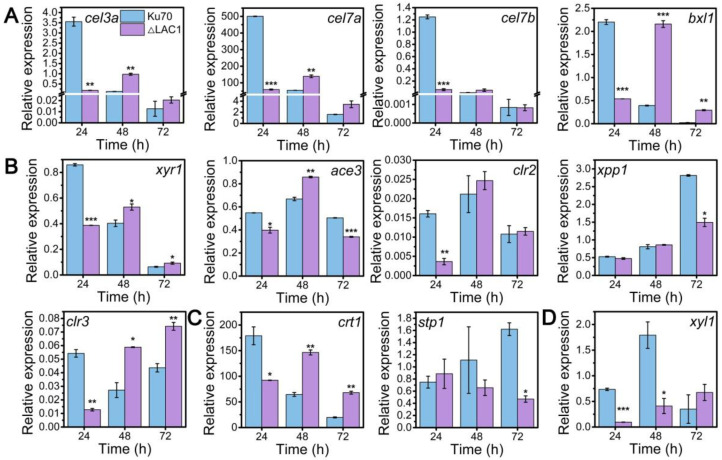
qRT-PCR analysis of the transcript abundance of genes relevant to cellulase and hemicellulase production in *T. reesei* Δku70 and Δlac1 grown on 2% lactose for 72 h including genes involved in (**A**) cellulase and hemicellulase production, (**B**) transcription factors, (**C**) sugar transportation, and (**D**) lactose metabolism. Values are the means of three biological replicates, and error bars are the standard deviations of these three replicates. The gene expression level was calculated using the 2^−ΔCt^ method. The relative mRNA level of each tested gene was normalized to the housekeeping gene *sar1*. * *p* < 0.05, ** *p* < 0.01, and *** *p* < 0.001.

**Figure 5 biomolecules-13-00295-f005:**
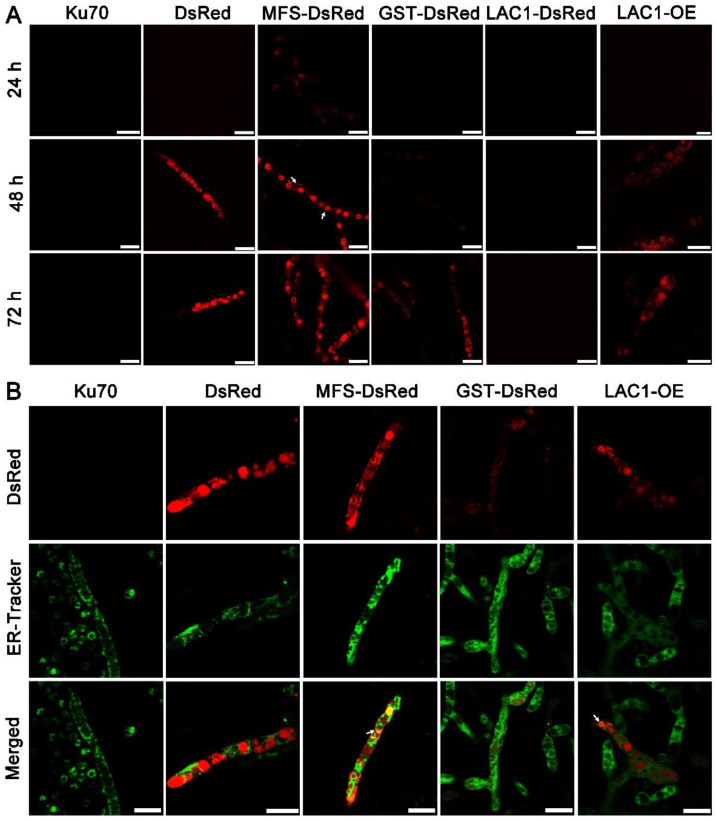
(**A**) Cellular localization of the three sugar transporters in *T. reesei* cultured in TMM + 2% lactose at 24 h, 48 h, and 72 h, respectively. The white arrow indicates the localization of MFS-DsRed on the cell membrane. (**B**) Confocal images of strains Δku70, DsRed, MFS-DsRed, GST-DsRed, LAC1-DsRed, and LAC1-DsRed-OE (LAC1-OE) with the treatment of ER-Tracker (green). Strains MFS-DsRed and GST-DsRed were constructed under their own endogenous promoters, and strain LAC1-DsRed-OE was constructed under the modified strong promoter cbh1.The white arrow points to the overlap of the green ER dye with the red fluorescent protein fusion LAC1-DsRed-OE. Scale bar = 10 μm.

**Figure 6 biomolecules-13-00295-f006:**
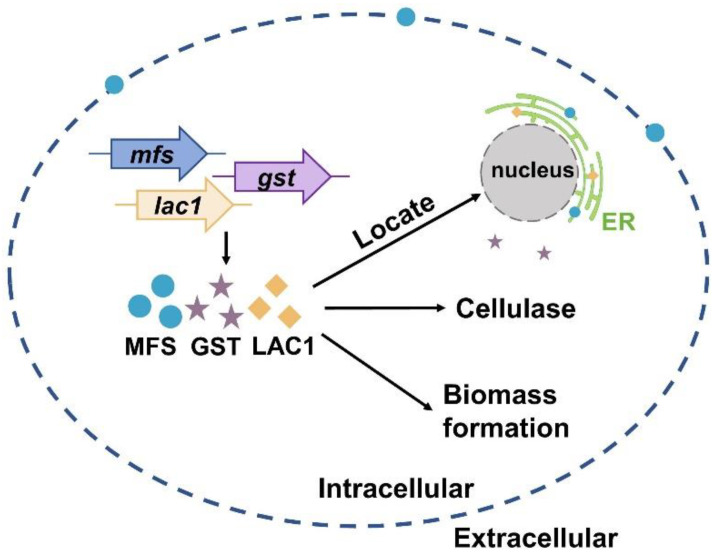
A scheme of the function and cellular distribution of three intracellular sugar transporters in *T. reesei* when cultured in lactose.

## Data Availability

All data are contained within the article and Appendix A.

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
