# Peer review of "Intracellular Sugar Transporters Facilitate Cellulase Synthesis in Trichoderma reesei Using Lactose"

_biomolecules, 2023, doi:10.3390/biom13020295_

Round 1
Reviewer 1 Report
The current manuscript describes the role of lactose sugar transporters in cellulase production from Trichoderma reesei. Actually, the research hypothesis revealed considerable novelty with strong supportive methodology, results, and discussion. However, some comments and hints should be considered before publications
General comments
1- The full name should be mentioned for the first time followed by an abbreviated short name in parentheses (for example (but not only) ER, kb in line 99 which should also be KB, PBS buffer, HBSS in line 133, FPase in line 170, etc……….)
2- Lines 116 and 125 after Table S1 add (supplementary material)
3- Section 2.3 is not clear and needs rephrasing. Additionally, the enzyme assay condition is not mentioned in references 18 or 19. Hence, enzymes assay should be included in the manuscript or related to the right references
4- Figures 3 and 4 are not cited in the manuscript text
5- In figure 3A: the fungal growth was measured through absorbance (a.u). could you elaborate on the applied protocol and included it in the revised manuscript?
6- The statistical analysis in figure 1 is irrational, especially in figure 1A
Minor and editing comments
Line 93: respectively need a comma
Line 94: it is Trichoderma minimal medium please correct
Line 102: comma before and after respectively
Line 103: it is Agrobacterium-mediated transformation please correct.
Line 132: it should be 570-700 nm, please correct.
Line 136: it should be 500-550 nm, please correct.
Line 138-139: rephrase this sentence
Line 145: containing RT-qPCR abbreviation, unify to qRT-PCR
Line 281: the parentheses shape is not compatible with the whole manuscripts
Author Response
Reviewer #1
General Comments: The current manuscript describes the role of lactose sugar transporters in cellulase production from Trichoderma reesei. Actually, the research hypothesis revealed considerable novelty with strong supportive methodology, results, and discussion. However, some comments and hints should be considered before publications.
Response: We want to thank the reviewer for stating that “Actually, the research hypothesis revealed considerable novelty with strong supportive methodology, results, and discussion.”. We are deeply grateful for the precious time, invaluable expertise, and superb professionalism the respected reviewer has put in improving the quality of our paper! We have made revisions according to the reviewers’ comments as mentioned below.
Comment 1: The full name should be mentioned for the first time followed by an abbreviated short name in parentheses (for example (but not only) ER, kb in line 99 which should also be KB, PBS buffer, HBSS in line 133, FPase in line 170, etc. ……….)
Response: We greatly appreciate this respected reviewer for raising this very professional comment! As suggested, the full name was mentioned for the first time followed by an abbreviated short name in parentheses in the whole manuscript, including the ones mentioned by the reviewer.
Comment 2: Lines 116 and 125 after Table S1 add (supplementary material).
Response: As suggested, we have added “(supplementary material)” after Table S1. (Line 145 and Line 167)
Comment 3: Section 2.3 is not clear and needs rephrasing. Additionally, the enzyme assay condition is not mentioned in references 18 or 19. Hence, enzymes assay should be included in the manuscript or related to the right references.
Response: As suggested, we have rephrased Section 2.3 as below.
“0.5 mL T. reesei conidia (107/mL) were seeded into 10 mL SDB and grown for 48 h at 28°C with 200 rpm. Then, 5 mL pre-grown mycelia (the wet weight of mycelia was about 0.6 g) were inoculated into a 250 mL conical flask containing 50 mL TMM media with 2% lactose and cultured for 72 h at 28℃ with 200 rpm. Samples were collected at different time points for cellulase and hemicellulase activity assay [20-24], confocal observation, and quantitative reverse transcription PCR (qRT-PCR) analysis. For analyzing the cell growth in culture medium, 1 mL fungal culture solution was collected and diluted appropriately and then the absorption value at 600 nm wavelength of diluent was measured using ultraviolet spectrophotometer (UV-2600, Shimadzu, Japan). In order to measure the growth ability of the recombinant strains on the solid medium, 10 μL of fresh spore solution was dropped to the center of the solid TMM plates. Then, the colony diameter was measured at 48 h and 72 h, respectively.” (Line 147-158)
Also, we are sorry for the inaccurate references. We have quoted new references in which the measurement methods of enzyme activities are described in detail. (Line 151)
Comment 4: Figures 3 and 4 are not cited in the manuscript text.
Response: We feel sorry for this mistake and we have cited Figure 3 and Figure 4 in the manuscript. (Line 336, 339, 342, 343, and 401)
Comment 5: In figure 3A: the fungal growth was measured through absorbance (a.u). could you elaborate on the applied protocol and included it in the revised manuscript?
Response: We sincerely appreciate this respected reviewer for raising this very important comment! We have elaborated on the applied protocol in the revised manuscript:
“For analyzing the cell growth in culture medium, 1 mL fungal culture solution was collected and diluted appropriately and then the absorption value at 600 nm wavelength of diluent was measured using ultraviolet spectrophotometer (UV-2600, Shimadzu, Japan).” (Line 152-155)
Comment 6: The statistical analysis in figure 1 is irrational, especially in figure 1A.
Response: We deeply appreciate this respected reviewer for raising this very critical comment! As suggested, we have corrected the significance analysis of the data in Figure 1. (Line 264)
Comment 7: Minor and editing comments:
Line 93: respectively need a comma.
Line 94: it is Trichoderma minimal medium please correct.
Line 102: comma before and after respectively.
Line 103: it is Agrobacterium-mediated transformation please correct.
Line 132: it should be 570-700 nm, please correct.
Line 136: it should be 500-550 nm, please correct.
Line 138-139: rephrase this sentence.
Line 145: containing RT-qPCR abbreviation, unify to qRT-PCR.
Line 281: the parentheses shape is not compatible with the whole manuscripts.
Response: As suggested, we have corrected all the above minor issues. Also, we have looked through the whole manuscript and tried our best to correct the typo errors we found.

Reviewer 2 Report
General comments:
Wang and coworkers report the discovery of intracellular sugar transporters that also assist on cellulase production in T. reesei. The claims were confirmed through different molecular biology techniques such as recombinant engineering, qRT-PCR analysis, deletion experiment, growth analysis, enzyme activity assay, and confocal imaging. Overall, the work is well done and results are interesting which will probably be useful for biotechnological purposes. Before the publication of this paper, I would like to recommend adding a figure (like illustration of the mechanism) showing how these intracellular sugar transporters function, how they catalyze or assist on hemicellulose degradation or cellulase production and their localization in the cell. The said figure would enable the readers to easily understand the discovery.
Author Response
General Comments: Wang and coworkers report the discovery of intracellular sugar transporters that also assist on cellulase production in T. reesei. The claims were confirmed through different molecular biology techniques such as recombinant engineering, qRT-PCR analysis, deletion experiment, growth analysis, enzyme activity assay, and confocal imaging. Overall, the work is well done and results are interesting which will probably be useful for biotechnological purposes. Before the publication of this paper, I would like to recommend adding a figure (like illustration of the mechanism) showing how these intracellular sugar transporters function, how they catalyze or assist on hemicellulose degradation or cellulase production and their localization in the cell. The said figure would enable the readers to easily understand the discovery.
Response: We thank the reviewer for stating that “The work is well done and results are interesting which will probably be useful for biotechnological purposes”. We are deeply grateful for the precious time, invaluable expertise, and superb professionalism the respected reviewer has put in improving the quality of our paper! We have made revisions according to the reviewers’ comments as mentioned below.
As suggested, we have made a scheme as Figure 6 showing the function and cellular distribution of the intracellular sugar transporters found in our study to make it easy to be understood by readers.
“In summary, we reported that sugar transporters MFS and LAC1 were localized in ER, while GST in cytoplasm. They have a positive role in cellulase production and biomass formation in T. reesei when cultured in lactose (Figure 6).” (Line 571-577)
Figure 6. A scheme of the function and cellular distribution of three intracellular sugar trans-porters in T. reesei when cultured in lactose.

Reviewer 3 Report
The work contributes to the knowledge of sugar transporters and cellulase regulation. The strategy is sound.
Descriptions must be reviewed in terms of use of English. Some examples:
l. 20-21 cellulase production when grown in lactose
l 21 mainly instead of majorly
59-60 have been identified as involved in...
84 offer new strategies to improve fungal cellulase production
99 to knock out genes....
115 primers are listed
l 121.. Clarify: why assay hemicellulase and not the other enzymes that form the cellulase complex? The activity of the other enzymes is described in section 3.2. No assay methods are cited.
Figure 1. The differences in scales might be misleading. Perhaps the figures could be placed in order of expression: crt, mfs....
l 165-175 you mean hemicellulase, not hemicellulose?
l 163 if you are referring to the genes the names should be written in italics and lowcase.
l 213 comparable to even higher....what does it mean?
l 225 C terminus
l 250 sugar transporters are ER-targeted
L 297 CRT1, both of which are essential....
l 323 lignocellulosic
Author Response
Reviewer #3
General Comments: The work contributes to the knowledge of sugar transporters and cellulase regulation. The strategy is sound. Descriptions must be reviewed in terms of use of English.
Response: We thank the reviewer for stating that “The work contributes to the knowledge of sugar transporters and cellulase regulation. The strategy is sound”. We are deeply grateful for the precious time, invaluable expertise, and superb professionalism the respected reviewer has put in improving the quality of our paper! We have extensively revised the manuscript in English and asked our English-speaking colleagues to help check it. We have made revisions according to the reviewers’ comments as mentioned below.
Comment 1: Line 20-21 cellulase production when grown in lactose.
Response: Revised as suggested. (Line 21)
Comment 2: Line 21 mainly instead of majorly.
Response: Revised as suggested. (Line 22)
Comment 3: Line 59-60 have been identified as involved in...
Response: Revised as suggested. (Line 70)
Comment 4: Line 84 offer new strategies to improve fungal cellulase production.
Response: Revised as suggested. (Line 95)
Comment 5: Line 99 to knock out genes .....
Response: Revised as suggested. (Line 127)
Comment 6: Line 115 primers are listed.
Response: Revised as suggested. (Line 145)
Comment 7: Line 121 Clarify: why assay hemicellulase and not the other enzymes that form the cellulase complex? The activity of the other enzymes is described in section 3.2. No assay methods are cited.
Response:Here, we use the parentheses to indicate cellulose or hemicellulose. To make it clearer, we corrected “(hemi)cellulase” to “cellulase and hemicellulase” in the whole manuscript. The literature cited for measuring enzyme activities is not accurate, so we have quoted new references in which the measurement methods of enzyme activities are described in detail. (Line 151)
Comment 8: Line Figure 1. The differences in scales might be misleading. Perhaps the figures could be placed in order of expression: crt, mfs.....
Response:As suggested, we have changed the order of the data in Figure 1, in the order of crt1, mfs, gst, and lac1. (Line 264-267)
Comment 9: Line 165-175, you mean hemicellulase, not hemicellulose?
Response: Here is a spelling mistake. What we want to express is hemicellulase. Therefore, we have changed “(hemi)cellulose” to “cellulase and hemicellulase”. (Line 274, 277, and 283)
Comment 10: Line 163 if you are referring to the genes the names should be written in italics and lowcase.
Response: We are mentioning these transporters at the protein level, so we wrote them in uppercase. (Line 272-273)
Comment 11: Line 213 comparable to even higher...., what does it mean?
Response: The conjunction "or" between “comparable to” and “even higher than” was missing, which has been added in the revised manuscript. So, the description should be “comparable to or even higher....,”. (Line 401)
Comment 12: Line 225 C terminus.
Response: Revised as suggested. (Line 433)
Comment 13: Line 250, sugar transporters are ER-targeted.
Response: Revised as suggested. (Line 465)
Comment 14: Line 297 CRT1, both of which are essential.....
Response: Revised as suggested. (Line 521)
Comment 15: Line 323 lignocellulosic.
Response: Revised as suggested. (Line 568)
